# The use of telemedicine to support Brazilian primary care physicians in managing eye conditions: The *TeleOftalmo* Project

Aline Lutz de Araujo[1,2]\*, Taís de Campos Moreira[3], Dimitris Rucks Varvaki Rados[1], Paula Blasco Gross[1], Cynthia Goulart Molina-Bastos[1], Natan Katz[1,4], Lisiane Hauser[1], Rodolfo Souza da Silva[1], Sabrina Dalbosco Gadenz[5], Rafael Gustavo Dal Moro[1], Felipe Cezar Cabral[3,4], Lucas Matturro[3], Cássia Garcia Moraes Pagano[3], Amanda Gomes Faria[3], Maicon Falavigna[3], Ana Célia da Silva Siqueira[1], Paulo Schor[2], Marcelo Rodrigues Gonçalves[1,6], Roberto Nunes Umpierre[1,6], Erno Harzheim[6,7]

1 Núcleo de Telessaúde, Universidade Federal do Rio Grande do Sul, Porto Alegre, Rio Grande do Sul, Brazil, 2 Departamento de Oftalmologia e Ciências Visuais, Universidade Federal de São Paulo, São Paulo, São Paulo, Brazil, 3 Hospital Moinhos de Vento, Porto Alegre, Rio Grande do Sul, Brazil, 4 Secretaria Municipal da Saúde, Prefeitura de Porto Alegre, Rio Grande do Sul, Brazil, 5 Hospital Sírio Libanês, São Paulo, São Paulo, Brazil, 6 Faculdade de Medicina, Universidade Federal do Rio Grande do Sul, Porto Alegre, Rio Grande do Sul, Brazil, 7 Ministério da Saúde, Brasília, Distrito Federal, Brazil

\* alinelutz.a@gmail.com

**Data Availability Statement:** All relevant data are within the manuscript and its Supporting Information files.

## Abstract

### Purpose

To determine whether teleophthalmology can help physicians in assessing and managing eye conditions and to ascertain which clinical conditions can be addressed by teleophthalmology in primary care setting.

### Methods

We evaluated the resolution capacity of *TeleOftalmo*, strategy implemented in the public health system of southern Brazil. Resolution capacity was defined as the ability to fully address patients' eye complaints in primary care with remote assistance from ophthalmologists. Data from tele-eye reports were collected over 14 months. Resolution capacity was compared across different age groups and different ocular conditions.

### Results

Overall, 8,142 patients had a tele-eye report issued in the study period. Resolution capacity was achieved in 5,748 (70.6%) patients. When stratified into age groups, the lowest capacity was 43.1% among subjects aged ≥65 years, while the highest was 89.7% among subjects aged 13–17 years (p<0.001). Refractive error (70.3%) and presbyopia (56.3%) were the most prevalent conditions followed by cataract (12.4%) and suspected glaucoma (7.6%). Resolution capacity was higher in cases of refractive error, presbyopia, spasm of accommodation and lid disorders than in patients diagnosed with other condition (p<0.001).

**Funding:** This work is based on research supported by Hospital Moinhos de Vento and the Program of Support for the Institutional Development of the Unified Health System (PROADI-SUS) (Website: http://www.saude.gov. br/acoes-e-programas/proadi-sus). The funders had no role in study design, data collection and analysis, decision to publish, or preparation of the manuscript.

**Competing interests:** The authors have declared that no competing interests exist.

## Conclusions

With telemedicine support, primary care physicians solved over two-thirds of patients' eye or vision complaints. Refractive errors had high case resolution rates, thus having a great impact on reducing the number of referrals to specialty care. Teleophthalmology adoption in primary-care settings as part of the workup of patients with eye or vision complaints promotes a more effective use of specialty centers and will hopefully reduce waiting times for specialty referral.

## Introduction

Worldwide, approximately 285 million people have impaired vision due to either eye diseases or uncorrected refractive errors (RE). [1] The latter alone affect 122 million people whose full vision could be restored with prescription glasses [1], 90% of whom live in low- and middle-income countries. [2] The World Health Organization has stated that individuals and families are frequently pushed into a cycle of poverty because of their inability to see well. [2]

In Brazil, uncorrected RE and cataract are the most common causes of impaired vision [3–5], followed by diabetic retinopathy and glaucoma. [5] These conditions can be effectively corrected or treated, and early detection may reduce disease burden and improve clinical outcomes. However, provision of access to secondary and tertiary care in the Brazilian public health system (*Sistema Único de Saúde*–SUS, Unified Health System) is still challenging. [6] Limited access results in long wait times for specialty care, underdiagnosis of eye conditions, and treatment delays.

*TeleOftalmo* is a telemedicine initiative for the Unified Health System that supports primary care with ophthalmologic diagnoses at a distance. Primary care physicians (PCP) can request a comprehensive "tele-eye exam" for their patients. Ophthalmologists working remotely examine patients' eyes in synchronous fashion, and then send a tele-eye report to the requesting PCP, who is in charge of prescriptions and other clinical management. The Brazilian legal framework does not allow physicians to provide prescriptions through telemedicine [7]; hence, PCPs play the central role in delivering direct care with the telemedicine support. Additional information on the *TeleOftalmo* project is presented in Box 1.

---

### Box 1 Main features of the *TeleOftalmo* project

• Conceived by TelessaúdeRS-UFRGS, a university telehealth center that creates and evaluates innovative telemedicine applications to overcome public healthcare challenges. [8]

• Headquarters located in the state capital, Porto Alegre, equipped with four private workstations for ophthalmologists;

• Eight remote units at primary or secondary care centers across the state;

• Trained nurses and technicians work at remote units under the supervision of a local general practitioner;

• Remote units and headquarters are networked through a high-speed, dedicated line;

• Current offer is 1,500–1,700 tele-eye exams per month;

• Each tele-eye exam covers visual acuity, refraction test, anterior segment imaging, non-dilated fundus examination, and intraocular pressure measurement;

• The ophthalmologist's report is sent to the Primary Care Physician through a secure, online platform;

• Main funding provided by the Brazilian Ministry of Health trough PROADI-SUS, a national program that sponsors Unified Health System development.

The purpose of this study is to determine whether the *TeleOftalmo* strategy can obviate the need for specialty appointments by solving common eye conditions in primary care facilities. The rationale of addressing eye problems by means of telemedicine at the primary level of care is to promote a more effective use of specialty centers and, expectedly, to reduce mean waiting time for those needing treatments provided only at the tertiary level.

## Materials and methods

### Settings

Rio Grande do Sul, with a population of about 11 million, is the southernmost state of Brazil. Fig 1 displays a map of the state with the distribution of *TeleOftalmo* headquarters and remote

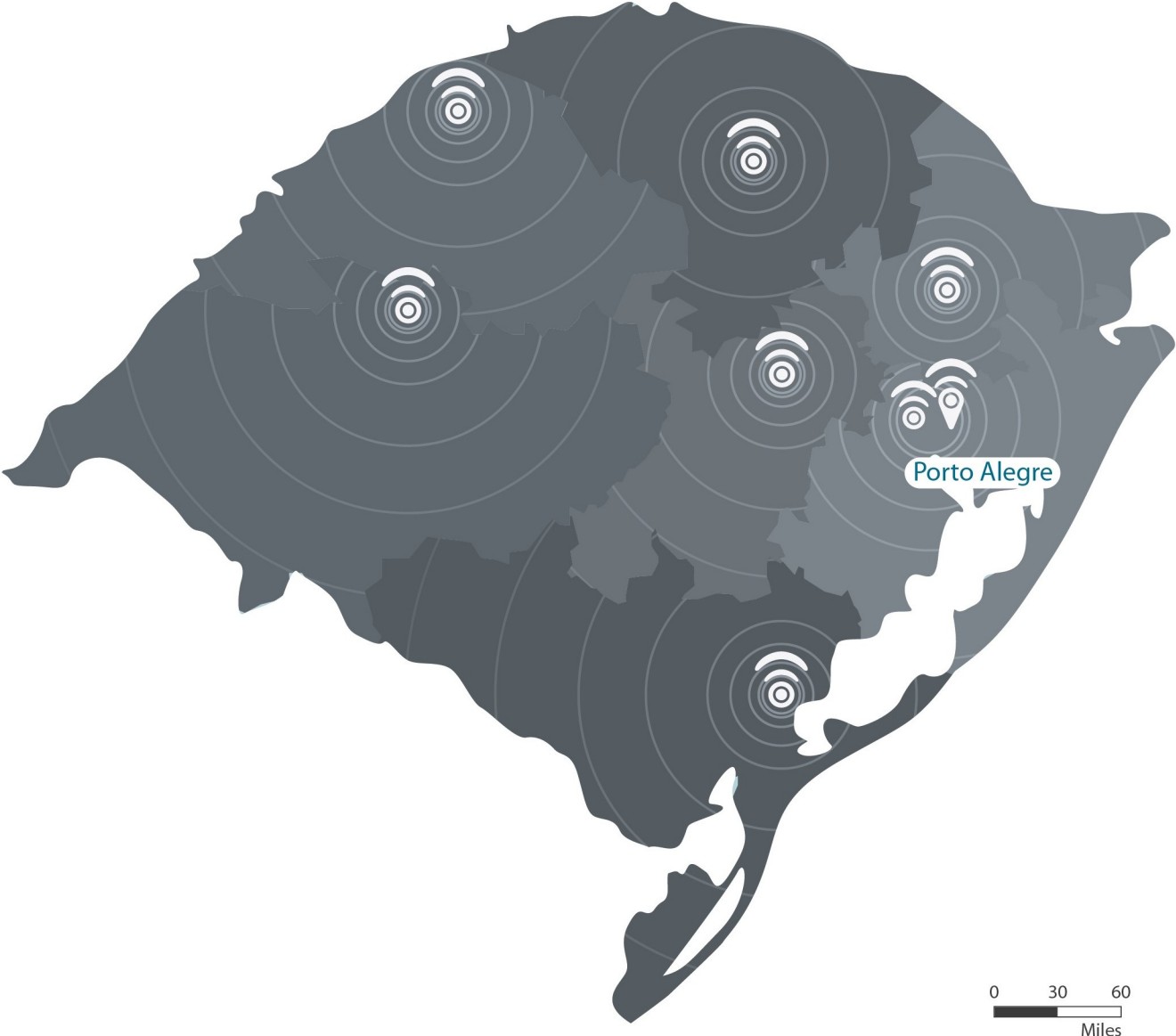

**Fig 1. Map of Rio Grande do Sul state.** *TeleOftalmo* units are located in the following municipalities: Porto Alegre (headquarters and two remote units), Farroupilha, Passo Fundo, Santa Rosa, Santiago, Santa Cruz do Sul, and Pelotas (one remote unit each).

units across the territory. The locations of remote units are defined according to population density and to existing health system regions and network.

## Patients

Patients over the age of 7 were eligible for *TeleOftalmo* if they presented to a primary care facility with one or more of the following: complaint of diminished vision; known or suspected RE; strabismus; disorders of the eyelids; conjunctival disorders; cataract without previous indication for surgery; or need for diabetic retinopathy screening. The exclusion criteria were: red eye or vision loss with acute onset; previously diagnosed eye disease requiring prompt medical or surgical treatment, such as uveitis, glaucoma, retinal disease, lacrimal drainage system disorder, orbital disease, and others; and cognitive impairment that would prevent adequate data collection.

## Study design and outcomes

We carried out a cross-sectional evaluation of all assessment reports issued during the first 14 months of *TeleOftalmo* operation.

The main outcome was the resolution capacity of the *TeleOftalmo* strategy, defined as the ability to fully address patients' eye complaints, under the following conditions:

1. The patient did not present with an eye condition requiring referral to a specialty center;

2. If the patient presented an eye condition, this condition was amenable to treatment and follow-up at the primary care setting, according to Rio Grande do Sul State Department of Health clinical protocols [9]; and

3. The respective tele-eye report did not recommend any further eye examination.

For purposes of analysis, patients were classified into the following age groups: children (age <12 years), teenagers (13–17 years), young adults (18–39 years), adults (40–64 years), and older adults (≥65 years). We also compared resolution capacity in diabetic and non-diabetic patients. Finally, we evaluated the prevalence of each telediagnosed eye condition and compared resolution capacity across conditions.

## The *TeleOftalmo* system

Our information technology (IT) team developed and implemented the following solutions for this project:

1. Telehealth platform: secure online platform that connects primary care services and the telehealth center (available at: www.plataformatelessaude.ufrgs.br). We developed a special module within this platform that includes electronic requests for telemedicine evaluations, scheduling, patient information, health data acquisition and storage, interpretation of data by consultant ophthalmologist, generation of tele-eye reports, and auditing activities. The module was specially designed to fulfill ophthalmology requirements; [10]

2. Telepresence system: videoconferencing system that uses remotely operated robotic cameras and live video streaming. The remote ophthalmologist controls all camera functions such as rotation and zoom. Camera's high resolution allows the remote ophthalmologist to see fine details in the patient's eyes, such as eye movements when testing extraocular motor function and pupillary size when testing pupillary reflexes. Live videoconferencing makes possible to perform interactive tests with patients such as subjective refraction;

3. Interoperability: The telehealth platform, a central Digital Imaging and Communication in Medicine (DICOM) server, and the diagnostic devices were integrated. All transfers of patient-identifying information from the platform to each device occurs through the server. Likewise, eye examination data acquired by the devices are readily stored in the server and made available at the consultant ophthalmologists' review workstations;

4. Medical equipment connected through a closed network: by means of networking remote units' equipment and ophthalmologists' workstations, we made possible to operate medical equipment from a distance. This includes digital refractors and visual acuity screens.

## Workflow

The PCP accesses the platform to request a teleophthalmology diagnostic. If the patient meets the enrollment criteria, the tele-eye exam is scheduled at the remote unit nearest the patient's home.

At the remote unit, nurses and technicians follow a detailed, predefined data acquisition workflow. The exams are: visual acuity; automatic refractometry; slit-lamp images of the eye's anterior segment; non-mydriatic retinography; and air-puff tonometry. Ophthalmologists working in the central unit evaluate the results immediately after data collection. All clinical information is displayed on private diagnostic workstations. Each workstation is equipped with three screens, so that the ophthalmologist can simultaneously see the video livestream and retrieve all acquired patient data, including numerical topometric data and ocular images (Fig 2). By means of a telepresence system equipped with high-resolution robotic cameras, the ophthalmologist performs real-time refraction tests and other examinations, such as inspection of the eyelids, tests of extraocular motor function, and pupillary reflexes. The specialist remotely controls a digital refractor and a visual acuity screen through a closed network to perform the refraction tests, to determine the need for corrective eyewear and the power of the corrective lens to be prescribed for the patient.

The ophthalmologist then issues a final report to the PCP through the platform. The report describes exam results, provides a diagnosis, and offers clinical recommendations. These recommendations are important for conditions that can be managed in the primary-care setting, such as RE, hordeolum, blepharitis, and mild dry eye. Should the patient need corrective

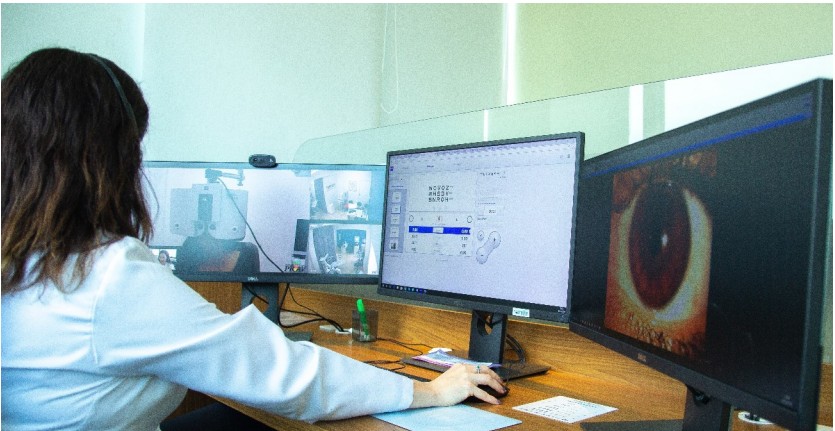

**Fig 2. Ophthalmologist interacting with patient through livestream video while performing a refraction test.** The screen on the right is displaying ocular images of the same patient.

glasses, they are provided free of charge within the project. Conversely, sight-threatening conditions prompt immediate referral; in such cases, the patient is given high priority for an in-person ophthalmological appointment. In this way, the PCP is responsible for decisions on how to manage the patient's case, supported by the ophthalmologist's report. During examinations at *TeleOftalmo*'s remote unit, the nursing staff and ophthalmologists do not provide, directly to patients, diagnoses or treatments necessary for their case.

## Statistical analysis

A descriptive analysis of patients' characteristics, complaints, and diagnoses was carried out. PASW Statistics, Version 18.0, was used for descriptive and statistical analyses. The chi-square test was used to compare proportions in each age and diagnosis group. Data are presented as numbers (percentage), except for age (mean ± standard deviation).

## Ethical aspects

The institutional review board of Hospital de Clínicas de Porto Alegre approved the research project. We obtained written consent from all patients or, if underaged, from parents or guardians.

## Results

During the study period, 8,142 patients completed a *TeleOftalmo* evaluation and had a corresponding report issued by the ophthalmologist to the PCP through the platform. Patient characteristics and demographics are presented in Table 1.

The overall resolution capacity of the *TeleOftalmo* strategy was 70.6% (5,748 of 8,142 patients). When stratified into age groups, capacity ranged from 43.1% (older adults) to 89.7% (teenagers). The proportion of solved cases among older adults was significantly lower than in the other age brackets (p<0.001). Fig 3 presents the results across all ages. Data of 33 individuals were missing and they were excluded from the analysis.

**Table 1. Demographic characteristics of the study population: The first 14 months of *TeleOftalmo* project (n = 8,142).**

| Age, years | Mean ± SD | 47.99 ± 19.03 |
|---|---|---|
| | Range | 8–96 |
| **Age groups** | Children (<12 years) | 428 (5.3) |
| No. (%) | Teenagers (13–17 years) | 497 (6.1) |
| | Young adults (18–39 years) | 1,336 (16.4) |
| | Adults (40–64 years) | 4,284 (52.6) |
| | Older adults (≥65 years) | 1,597 (19.6) |
| **Gender** | Female | 5,497 (67.5) |
| No. (%) | Male | 2,645 (35.5) |
| **Ethnicity** | European | 6,290 (77.6) |
| No. (%) | African | 545 (6.7) |
| | Indigenous | 16 (0.2) |
| | Asian | 29 (0.4) |
| | Mixed | 1,259 (15.5) |
| **Diabetes** | Yes | 1,157 (14.2) |
| No. (%) | No | 6,985 (85.8) |

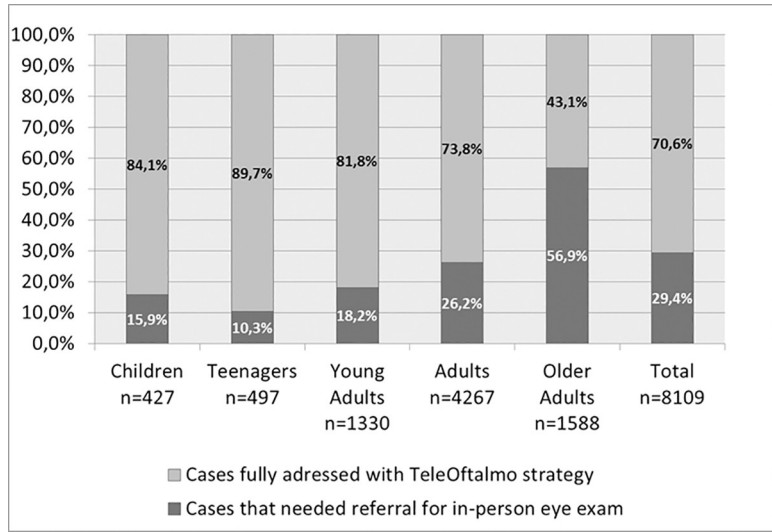

**Fig 3. Resolution capacity of the *TeleOftalmo* strategy across different age groups (p<0,001).** Pearson's chi square adjusted residuals indicated higher proportion of resolution for children, teenagers, young adults, and adults; and lower proportion of resolution for older adults. Numbers in bars express percentage.

The single most prevalent condition was any RE (myopia, astigmatism, and hyperopia), which accounted for 70.3% of all diagnoses. Table 2 presents the resolution capacity according to reported ocular conditions. Resolution capacity was significantly higher in cases of RE, presbyopia, spasm of accommodation and lid disorders than in patients diagnosed with any other ocular condition (p<0.001). Patients with diabetic retinopathy, retinal vascular occlusion, and glaucoma or glaucoma suspect needed ophthalmology referral frequently.

Diabetic patients accounted for 14.2% of the population (1,157 of 8,142 patients). Resolution capacity was 53.1% among diabetic patients and 73.5% among non-diabetic patients (p<0.001), explained by a higher prevalence of eye conditions that required referral among diabetic patients: cataracts (20.5% in diabetics vs. 11.1% in non-diabetics, p<0.001), diabetic retinopathy (17.0% vs. 0.7%, p<0.001), and suspected glaucoma (11.0% vs. 7.0%, p<0.001).

**Table 2. Resolution capacity of the *TeleOftalmo* strategy across diagnoses from the tele-eye reports.** Pearson's chi square adjusted residuals indicated higher (♦) and lower (*) proportion of resolution for each condition, except for hypertensive retinopathy. Each patient might have more than one diagnosis.

| Diagnosis | Number of diagnosis | Prevalence | Resolution capacity | P (Chi-square) |
|---|---|---|---|---|
| Refractive error | 5710 | 70,3% | 76,5% | <0,001 ♦ |
| Presbyopia | 4571 | 56,3% | 75,4% | <0,001 ♦ |
| Strabismus | 82 | 1,0% | 43,2% | <0,001 * |
| Spasm of accommodation | 51 | 0,6% | 90,2% | 0,002 ♦ |
| Lid disorders | 540 | 6,6% | 77,6% | <0,001 ♦ |
| Conjunctival disorders | 382 | 4,7% | 56,3% | <0,001 * |
| Cataracts | 1011 | 12,4% | 24,9% | <0,001 * |
| Suspected glaucoma | 617 | 7,6% | 7,7% | <0,001 * |
| Glaucoma | 100 | 1,2% | 6,0% | <0,001 * |
| Nonproliferative diabetic retinopathy | 244 | 3,0% | 21,7% | <0,001 * |
| Proliferative diabetic retinopathy | 50 | 0,6% | 12,0% | <0,001 * |
| Hypertensive retinopathy | 91 | 1,1% | 62,2% | 0,08 |
| Retinal vascular occlusion | 23 | 0,3% | 4,3% | <0,001 * |
| Others | 1486 | 18,3% | 55,4% | 0,001 * |

## Discussion

In our series, resolution capacity was around 70%; that is, 70% of all patients that would have been placed on a waiting list for ophthalmologist referral without the *TeleOftalmo* strategy were successfully managed in primary care instead. Consequently, we expect to have a significant impact on waiting times for ophthalmological treatment as the project scales up. Attempts to reduce waiting times with the use of telemedicine have already shown good results in Scotland. [11] Following the recommendations of the United Kingdom short waiting-time guidelines for outpatient appointments, the referral pathway was modified with the use of a fully electronic ophthalmic-referral service with digital imaging. The use of a centralized unit that interprets eye images for clinical decision-making prevented the need for appointment in 37% of referrals. [11] Our resolution capacity was higher than that of this study, probably due to the fact that we offered a comprehensive eye exam rather than eye imaging only. Additionally, it is important to note that our tele-eye exam aims at patients that do not have a clear indication for referral; patients already diagnosed with any eye disease that requires treatment are directly referred to tertiary centers, as described in our exclusion criteria. The selection criteria imply that the resolution capacity observed applies only to the population enrolled in the project and would not be as high if the whole waiting list for ophthalmology care were considered.

The lowest resolution capacity was found among older adults (age ≥65 years). Ocular aging is a known risk factor for several diseases, such as cataract, glaucoma, and age-related macular degeneration. [12,13] As expected, the prevalence of these conditions was higher among older people in our sample. The advantages of diagnosing conditions that cannot be treated in primary care are to enable straightforward referrals to specialty centers (for example, cataract patients to cataract surgery centers) and to prioritize more severe cases on the waiting list. In some conditions, establishing the diagnosis ahead of a conventional in-person visit may also allow the patient to benefit from modification of risk factors, such as improving glycemic control in diabetic patients once they are diagnosed with diabetic retinopathy.

In this study, RE was the most prevalent condition. This is consistent with the literature, in which refraction problems and presbyopia are the main reported reasons for seeking eye care. [14] Uncorrected RE is a major cause of moderate to severe vision impairment in low- and middle-income regions. [15] Nevertheless, greater emphasis has historically been placed on eye diseases rather than on refractive problems, both in health policy and in research. [16,17] Within the *TeleOftalmo* paradigm, refraction tests are part of telemedicine-based comprehensive eye exams offered in primary care units. This allows patients to remain in the community and avoids long waits for specialty referral to address visual deficits due to simple RE. It bears stressing that, in Brazil, ophthalmologists are the professionals in charge of performing refraction tests and prescribing corrective eyewear. Ophthalmological services are highly concentrated in tertiary care centers, unevenly distributed across the country, and any eye complaint would require referral from primary care to a specialist. Patients with RE are placed on the same waiting list as those with undiagnosed, progressive eye diseases. The aim of *TeleOftalmo* was to alleviate the large, growing demand for specialist ophthalmologist care and to diagnose eye conditions sooner than if patients had waited for an in-person visit to a specialist. As many eye conditions can be treated in primary care once properly diagnosed, it is well established that consultant ophthalmologists can assist primary care physicians through telemedicine. [18–20] Second, this strategy fast-tracks patients with sight-threatening diseases that would otherwise have their treatment delayed. Glaucoma and retinal diseases, for instance, may lead to permanent vision loss without timely treatment.

To our knowledge, this is the first teleophthalmology program in the world that performs real-time subjective refraction, and the first one in Brazil that offers comprehensive eye exams.

Most of the prior initiatives have focused on a model of forwarding fundus photographs for remote reading, with effective results for detecting some specific diseases. [21–23] These initiatives are usually intended to screen a specific disease in a high-risk population–for instance, diabetic retinopathy in diabetic patients [24,25] and retinopathy of prematurity in premature newborns. [26,27] The *TeleOftalmo* project, on the other hand, is dedicated to comprehensive eye-checks and providing glass prescription. This is particularly important in our setting due to unmet demand for ophthalmological visits, as discussed above.

Adequate IT infrastructure is critical for telemedicine. To achieve *TeleOftalmo*'s purposes, it was necessary to develop specific workflows and an integrated data system that covered the entire pathway from the tele-eye request to the final report. By the time the project was implemented, there was no off-the-shelf system that would provide the adequate networking between the requesting PCP, the assistant team that verifies if the patient meets the study criteria and schedules the exams, the nursing team at the remote units, and the remote ophthalmologists. The system not only had to manage health data securely, but it also should follow the clinical steps we planned, which includes an initial asynchronous data acquisition, followed by medical review and hypothesis formulation, synchronous medical examination and final tele-eye report. By developing our own telehealth platform, these needs were met.

Interoperability of medical equipment and systems is another issue we tackled. Although there have been significant improvements in healthcare IT, it is still a complex and intricate task to integrate practice management system, electronic health records (EHR), and eye care equipment.[28] Interoperability represents the ability of systems and devices to easily exchange data, regardless of the vendor or brand. [28] Since most ophthalmic device outputs are still in a proprietary format and cannot be readily imported into an EHR [10], it has been a central part of our project to overcome these issues. Interoperability minimizes the risk of patient misidentification and reduces technicians' working time, since repetitive data entry is no longer necessary. [29] It may also contribute to the remote ophthalmologist's productivity, given that all images and test results are readily available from the same server and retrieved in one single viewer.

Other experiences with teleophthalmology reported in the literature have used the primary care setting as well, in order to provide patients with the opportunity to receive eye care via telemedicine as part of their routine visit. [30] In Atlanta, Georgia, a program called Technology-Based Eye Care Services (TECS) was established in 5 primary care clinics, where technicians were trained to collect information about the patient's eyes. [31] The information then is interpreted remotely, and patients with possible abnormal findings are scheduled for a face-to-face examination. TECS and TeleOftalmo share many common features: they are teleophthalmological services based in primary care facilities, provide comprehensive eye-checks, and ocular diseases prompt an in-person exam. However, *TeleOftalmo* makes use of real-time virtual encounters between patient and ophthalmologist. These encounters allow the ophthalmologist to interact with the patient and perform subjective refraction. Based on trial and error and patient collaboration, subjective refraction warrants the best refinements in the prescription of glasses. Other tests are possible to be performed through our high-resolution robotic cameras, such as inspection of the eyelid position and function, extraocular motor function, and pupillary reflexes. Therefore, the differential of *TeleOftalmo* is the use of real-time interactive telemedicine and remotely operated equipment. These features may have contributed to the case-solving capacity reported in this study.

## Conclusion

We report the results of a novel telemedicine project that supplements primary care with tele-diagnosis in ophthalmology in a limited-resource country. Offering diagnostic through

telemedicine allows more precise identification of patients with sight-threatening diseases, who need prompt referral to an ophthalmologist and, conversely, cases that are well suited for primary care management. Over two-thirds of our sample could be managed at the primary care level with the aid of a remote consultant ophthalmologist. The strategy had a higher resolution capacity for patients under the age of 65 who presented with refractive error and/or presbyopia. By incorporating regular use of teleophthalmology in the primary care setting as part of the workup of patients with eye complaints, we expect to achieve a more effective use of specialty centers, whose limited capacity enforces a limit on the number of patients that can be seen in person.

## Author Contributions

**Conceptualization:** Aline Lutz de Araujo, Taís de Campos Moreira, Felipe Cezar Cabral, Lucas Matturro, Maicon Falavigna, Paulo Schor, Roberto Nunes Umpierre, Erno Harzheim.

**Data curation:** Sabrina Dalbosco Gadenz, Maicon Falavigna.

**Formal analysis:** Aline Lutz de Araujo, Lisiane Hauser, Maicon Falavigna.

**Funding acquisition:** Taís de Campos Moreira, Felipe Cezar Cabral.

**Investigation:** Aline Lutz de Araujo, Taís de Campos Moreira, Sabrina Dalbosco Gadenz, Lucas Matturro, Amanda Gomes Faria, Maicon Falavigna, Marcelo Rodrigues Gonçalves, Roberto Nunes Umpierre.

**Methodology:** Aline Lutz de Araujo, Taís de Campos Moreira, Dimitris Rucks Varvaki Rados, Natan Katz, Lisiane Hauser, Sabrina Dalbosco Gadenz, Felipe Cezar Cabral, Lucas Matturro, Cássia Garcia Moraes Pagano, Amanda Gomes Faria, Maicon Falavigna, Paulo Schor, Marcelo Rodrigues Gonçalves, Roberto Nunes Umpierre, Erno Harzheim.

**Project administration:** Aline Lutz de Araujo, Taís de Campos Moreira, Dimitris Rucks Varvaki Rados, Natan Katz, Sabrina Dalbosco Gadenz, Lucas Matturro, Cássia Garcia Moraes Pagano, Ana Célia da Silva Siqueira, Marcelo Rodrigues Gonçalves.

**Software:** Rafael Gustavo Dal Moro.

**Supervision:** Aline Lutz de Araujo, Taís de Campos Moreira, Paula Blasco Gross, Sabrina Dalbosco Gadenz, Rafael Gustavo Dal Moro, Felipe Cezar Cabral, Lucas Matturro, Cássia Garcia Moraes Pagano, Amanda Gomes Faria.

**Validation:** Aline Lutz de Araujo, Taís de Campos Moreira, Dimitris Rucks Varvaki Rados, Paula Blasco Gross, Cynthia Goulart Molina-Bastos, Sabrina Dalbosco Gadenz, Rafael Gustavo Dal Moro, Lucas Matturro.

**Writing – original draft:** Aline Lutz de Araujo, Taís de Campos Moreira, Dimitris Rucks Varvaki Rados, Paula Blasco Gross, Natan Katz, Lisiane Hauser, Sabrina Dalbosco Gadenz, Felipe Cezar Cabral, Lucas Matturro, Cássia Garcia Moraes Pagano, Maicon Falavigna, Ana Célia da Silva Siqueira, Marcelo Rodrigues Gonçalves, Erno Harzheim.

**Writing – review & editing:** Aline Lutz de Araujo, Taís de Campos Moreira, Dimitris Rucks Varvaki Rados, Paula Blasco Gross, Cynthia Goulart Molina-Bastos, Natan Katz, Rodolfo Souza da Silva, Sabrina Dalbosco Gadenz, Felipe Cezar Cabral, Lucas Matturro, Cássia Garcia Moraes Pagano, Amanda Gomes Faria, Maicon Falavigna, Paulo Schor, Marcelo Rodrigues Gonçalves, Roberto Nunes Umpierre, Erno Harzheim.

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
