## [Decision Letter · Decision Letter 0]

22 Oct 2019

PONE-D-19-24060

The use of telemedicine to support Brazilian primary care physicians in managing eye conditions: the TeleOftalmo Project

PLOS ONE

Dear Dr MOREIRA,

Thank you for submitting your manuscript to PLOS ONE. After careful consideration, we feel that it has merit but does not fully meet PLOS ONE’s publication criteria as it currently stands. Therefore, we invite you to submit a revised version of the manuscript that addresses the points raised during the review process.

Please answer for both reviewers’ and take extensive care for the methodological contribution that

should be emphasized more with highlighting any clinical/technological (IT) considerations

that makes the system competitive or superior comparing with other solutions.

We would appreciate receiving your revised manuscript by Dec 06 2019 11:59PM. To enhance the reproducibility of your results, we recommend that if applicable you deposit your laboratory protocols in protocols.io, where a protocol can be assigned its own identifier (DOI) such that it can be cited independently in the future. For instructions see: http://journals.plos.org/plosone/s/submission-guidelines#loc-laboratory-protocols

We look forward to receiving your revised manuscript.

Kind regards,

Adrienne Csutak, MD, PhD, MSc

Academic Editor

PLOS ONE

**Journal Requirements:**

2. Please state in your methods section whether you obtained consent from parents or guardians of the minors included in the study or whether the research ethics committee or IRB approved the lack of parent or guardian consent.

3. We note that  Figure(s) 1 in your submission contain [map/satellite] images which may be copyrighted. All PLOS content is published under the Creative Commons Attribution License (CC BY 4.0), which means that the manuscript, images, and Supporting Information files will be freely available online, and any third party is permitted to access, download, copy, distribute, and use these materials in any way, even commercially, with proper attribution. For these reasons, we cannot publish previously copyrighted maps or satellite images created using proprietary data, such as Google software (Google Maps, Street View, and Earth). For more information, see our copyright guidelines: http://journals.plos.org/plosone/s/licenses-and-copyright.

a) You may seek permission from the original copyright holder of Figure(s) [#] to publish the content specifically under the CC BY 4.0 license.  

**Comments to the Author**

1. Is the manuscript technically sound, and do the data support the conclusions?

Reviewer #1: Yes

Reviewer #2: Partly

2. Has the statistical analysis been performed appropriately and rigorously? 

Reviewer #1: Yes

Reviewer #2: Yes

3. Have the authors made all data underlying the findings in their manuscript fully available?

Reviewer #1: Yes

Reviewer #2: Yes

4. Is the manuscript presented in an intelligible fashion and written in standard English?

Reviewer #1: Yes

Reviewer #2: Yes

5. Review Comments to the Author

Reviewer #1: Dear Authors,

In my opinion, the paper is well written and contributes to the existing knowledge. I could not find any logical errors in the presentation and the approaches used.

I have provided my comments/questions for consideration:

1. The p-values should be linked to presented tables to visualize significance (e.g.: new column for p-value or 95%CI).

2. Were assumptions fulfilled in case of Chi-squared test for each strata? (at least 5 obs in 80% of all cells)

Regards,

Reviewer #2: This paper presents experimental results regarding a teleophthalmology system

implemented in the public health system of southern Brazil. The aim of the work

was to measure up the ability of the system to fully address patients’ eye complaints

in primary care with remote assistance from ophthalmologists.

The purpose of the action is clear and currently an important field; the manuscript

is well-written and easy to follow from these aspects. The study is well-prepared and

the evaluation has been performed convincingly with some positive outcome regarding

the benefits of applying the system in health care.

In general, I think that the results are reasonable and interesting for a wide

-- also scientific -- audience. My greatest concern that should be resolved to

propose the publication of the paper is a much deeper comparison with other

existing teleophthalmology systems; some such effort has been made in the manuscript,

but I definitely suggest widening this study. Especially, the methodological contribution

could be emphasized more with highlighting any clinical/technological (IT) considerations

that makes the system competitive or superior comparing with other solutions.

6. PLOS authors have the option to publish the peer review history of their article (what does this mean?). If published, this will include your full peer review and any attached files.

Reviewer #1: No

Reviewer #2: No

---

## [Author Response · Author response to Decision Letter 0]

16 Nov 2019

Journal Requirements: 

a. A rebuttal letter that responds to each point raised by the academic editor and reviewer(s). This letter should be uploaded as separate file and labeled 'Response to Reviewers'.

b. A marked-up copy of your manuscript that highlights changes made to the original version. This file should be uploaded as separate file and labeled 'Revised Manuscript with Track Changes'.

c. An unmarked version of your revised paper without tracked changes. This file should be uploaded as separate file and labeled 'Manuscript'.

Response: These files were provided.

2) Please state in your methods section whether you obtained consent from parents or guardians of the minors included in the study or whether the research ethics committee or IRB approved the lack of parent or guardian consent.

Response: We have now included the information in Methods. 

3) We note that Figure(s) 1 in your submission contain [map/satellite] images which may be copyrighted. All PLOS content is published under the Creative Commons Attribution License (CC BY 4.0), which means that the manuscript, images, and Supporting Information files will be freely available online, and any third party is permitted to access, download, copy, distribute, and use these materials in any way, even commercially, with proper attribution. For these reasons, we cannot publish previously copyrighted maps or satellite images created using proprietary data, such as Google software (Google Maps, Street View, and Earth).For these reasons, we cannot publish previously copyrighted maps or satellite images created using proprietary data, such as Google software (Google Maps, Street View, and Earth). 

 We require you to either (1) present written permission from the copyright holder to publish these figures specifically under the CC BY 4.0 license, or (2) remove the figures from your submission.

Response: Figure 1 contains a map that was designed by our Department's designers, based on publicly available maps from Brazil. They did not use proprietary data or softwares, such as Google Maps. As such, we would keep the figure, but if the Editor recommends otherwise, we can remove it.

Comments to the Author

Reviewer #1: 

Dear Authors,

In my opinion, the paper is well written and contributes to the existing knowledge. I could not find any logical errors in the presentation and the approaches used.

I have provided my comments/questions for consideration:

1. The p-values should be linked to presented tables to visualize significance (e.g.: new column for p-value or 95%CI).

Response: We thank you for the opportunity to review this point. We added the p-values in table 2 and figure 3. Regarding the table 2, after reviewing the manuscript, we have changed its presentation to provide more information on resolution capacity of the teleophthalmology program. 

The previous table 2 was intended to describe the prevalence of diagnoses across all age groups, but not to directly compare them. If the editors and/or reviewers still find informative to present this table, we can paste it as supplementary material. 

2. Were assumptions fulfilled in case of Chi-squared test for each strata? (at least 5 obs in 80% of all cells).

Response: Yes, they were. As mentioned above, we changed the presentation of results and now Chi-square results (including p-values and adjusted residuals) are more explicit.

Reviewer #2: This paper presents experimental results regarding a teleophthalmology system implemented in the public health system of southern Brazil. The aim of the work was to measure up the ability of the system to fully address patients’ eye complaints in primary care with remote assistance from ophthalmologists.

The purpose of the action is clear and currently an important field; the manuscript is well-written and easy to follow from these aspects. The study is well-prepared and the evaluation has been performed convincingly with some positive outcome regarding the benefits of applying the system in health care.

In general, I think that the results are reasonable and interesting for a wide -- also scientific -- audience. My greatest concern that should be resolved to propose the publication of the paper is a much deeper comparison with other

existing teleophthalmology systems; some such effort has been made in the manuscript, but I definitely suggest widening this study. Especially, the methodological contribution could be emphasized more with highlighting any clinical/technological (IT) considerations that makes the system competitive or superior comparing with other solutions.

Response: Thank you for giving us the opportunity to increase the quality of our report. We included a detailed description of IT developments for this project in the Methods section, under The TeleOftalmo System subsection. In Discussion, we now have explored these developments and the associated clinical workflow in-depth. We also have compared our work to other primary-care based teleophthalmological programs - in summary, the differential of TeleOftalmo is that we offer a comprehensive eye exam, instead of screening only for specific diseases, and that we use real-time interactive telemedicine and remotely operated equipment.

---

## [Decision Letter · Decision Letter 1]

10 Feb 2020

PONE-D-19-24060R1

The use of telemedicine to support Brazilian primary care physicians in managing eye conditions: the TeleOftalmo Project

PLOS ONE

Dear Dr MOREIRA,

Thank you for submitting your manuscript to PLOS ONE. After careful consideration, we feel that it has merit but does not fully meet PLOS ONE’s publication criteria as it currently stands. Therefore, we invite you to submit a revised version of the manuscript that addresses the points raised during the review process.

Please make the requested minor modification and than we would appreciate receiving your revised manuscript by Mar 26 2020 11:59PM. To enhance the reproducibility of your results, we recommend that if applicable you deposit your laboratory protocols in protocols.io, where a protocol can be assigned its own identifier (DOI) such that it can be cited independently in the future. For instructions see: http://journals.plos.org/plosone/s/submission-guidelines#loc-laboratory-protocols

We look forward to receiving your revised manuscript.

Kind regards,

Adrienne Csutak, MD, PhD, MSc

Academic Editor

PLOS ONE

Reviewers' comments:

Reviewer's Responses to Questions

**Comments to the Author**

1. If the authors have adequately addressed your comments raised in a previous round of review and you feel that this manuscript is now acceptable for publication, you may indicate that here to bypass the “Comments to the Author” section, enter your conflict of interest statement in the “Confidential to Editor” section, and submit your "Accept" recommendation.

Reviewer #3: (No Response)

2. Is the manuscript technically sound, and do the data support the conclusions?

Reviewer #3: Yes

3. Has the statistical analysis been performed appropriately and rigorously? 

Reviewer #3: Yes

4. Have the authors made all data underlying the findings in their manuscript fully available?

Reviewer #3: Yes

5. Is the manuscript presented in an intelligible fashion and written in standard English?

Reviewer #3: Yes

6. Review Comments to the Author

Reviewer #3: The manuscript describes a study of a telemedicine application to ophthalmology. The purpose was to show the extent that tele-ophthalmology as practiced by the authors can distinguish between cases requiring referral to an ophthalmologist versus cases that can be managed at the primary care level, which potentially is of interest to communities with certain limited resources.

The authors have responded properly to the previous reviewers’ comments and questions. In the present form of the manuscript, the treatment of the data is appropriate, the amount of detail is sufficient, and the figures and tables are acceptable. However, in the originally submitted manuscript, there was a line item in Table 2 for ‘accommodation disorder’ that is absent in the revised Table 2. The authors should correct this omission before publication.

7. PLOS authors have the option to publish the peer review history of their article (what does this mean?). If published, this will include your full peer review and any attached files.

Reviewer #3: Yes: David M Silver

---

## [Author Response · Author response to Decision Letter 1]

4 Mar 2020

6. Review Comments to the Author

Reviewer #3: The manuscript describes a study of a telemedicine application to ophthalmology. The purpose was to show the extent that tele-ophthalmology as practiced by the authors can distinguish between cases requiring referral to an ophthalmologist versus cases that can be managed at the primary care level, which potentially is of interest to communities with certain limited resources.

The authors have responded properly to the previous reviewers’ comments and questions. In the present form of the manuscript, the treatment of the data is appropriate, the amount of detail is sufficient, and the figures and tables are acceptable. However, in the originally submitted manuscript, there was a line item in Table 2 for ‘accommodation disorder’ that is absent in the revised Table 2. The authors should correct this omission before publication.

RESPONSE:

 Thank you for catching this missing condition. We have now added the missing condition to Table 2, under the name “Spasm of accommodation“. The term “spasm of accommodation” was used instead of “accommodation disorder” in order not to be confused with presbyopia.

---

## [Editor Report · Decision Letter 2]

16 Mar 2020

The use of telemedicine to support Brazilian primary care physicians in managing eye conditions: the TeleOftalmo Project

PONE-D-19-24060R2

Dear Dr. MOREIRA,

We are pleased to inform you that your manuscript has been judged scientifically suitable for publication and will be formally accepted for publication once it complies with all outstanding technical requirements.

With kind regards,

Adrienne Csutak, MD, PhD, MSc

Academic Editor

PLOS ONE

---

## [Editor Report · Acceptance letter]

20 Mar 2020

PONE-D-19-24060R2 

The use of telemedicine to support Brazilian primary care physicians in managing eye conditions: the *TeleOftalmo* Project 

Dear Dr. MOREIRA:

I am pleased to inform you that your manuscript has been deemed suitable for publication in PLOS ONE. Congratulations! Your manuscript is now with our production department. 

With kind regards,

on behalf of

Dr. Adrienne Csutak 

Academic Editor

PLOS ONE